# Novel Epoxides of Soloxolone Methyl: An Effect of the Formation of Oxirane Ring and Stereoisomerism on Cytotoxic Profile, Anti-Metastatic and Anti-Inflammatory Activities In Vitro and In Vivo

**DOI:** 10.3390/ijms23116214

**Published:** 2022-06-01

**Authors:** Oksana V. Salomatina, Aleksandra V. Sen’kova, Arseny D. Moralev, Innokenty A. Savin, Nina I. Komarova, Nariman F. Salakhutdinov, Marina A. Zenkova, Andrey V. Markov

**Affiliations:** 1Institute of Chemical Biology and Fundamental Medicine, Siberian Branch of the Russian Academy of Sciences, Lavrent’ev Avenue, 8, 630090 Novosibirsk, Russia; ana@nioch.nsc.ru (O.V.S.); senkova_av@niboch.nsc.ru (A.V.S.); a.moralev@g.nsu.ru (A.D.M.); savin_ia@niboch.nsc.ru (I.A.S.); marzen@niboch.nsc.ru (M.A.Z.); 2N.N. Vorozhtsov Novosibirsk Institute of Organic Chemistry, Siberian Branch of the Russian Academy of Sciences, Lavrent’ev Avenue, 9, 630090 Novosibirsk, Russia; komar@nioch.nsc.ru (N.I.K.); anvar@nioch.nsc.ru (N.F.S.); 3Faculty of Natural Sciences, Novosibirsk State University, Pirogova Str., 1, 630090 Novosibirsk, Russia

**Keywords:** soloxolone methyl, epoxide, pharmacophore group, apoptosis, anti-metastatic activity, anti-inflammatory activity, toxicity, 18βH-glycyrrhetinic acid, ABC transporters, CDDO-Me

## Abstract

It is known that epoxide-bearing compounds display pronounced pharmacological activities, and the epoxidation of natural metabolites can be a promising strategy to improve their bioactivity. Here, we report the design, synthesis and evaluation of biological properties of **αO-SM** and **βO-SM**, novel epoxides of soloxolone methyl (**SM**), a cyanoenone-bearing derivative of 18βH-glycyrrhetinic acid. We demonstrated that the replacement of a double-bound within the cyanoenone pharmacophore group of **SM** with α- and β-epoxide moieties did not abrogate the high antitumor and anti-inflammatory potentials of the triterpenoid. It was found that novel **SM** epoxides induced the death of tumor cells at low micromolar concentrations (IC_50_^(24h)^ = 0.7–4.1 µM) via the induction of mitochondrial-mediated apoptosis, reinforced intracellular accumulation of doxorubicin in B16 melanoma cells, probably by direct interaction with key drug efflux pumps (P-glycoprotein, MRP1, MXR1), and the suppressed pro-metastatic phenotype of B16 cells, effectively inhibiting their metastasis in a murine model. Moreover, **αO-SM** and **βO-SM** hampered macrophage functionality in vitro (motility, NO production) and significantly suppressed carrageenan-induced peritonitis in vivo. Furthermore, the effect of the stereoisomerism of **SM** epoxides on the mentioned bioactivities and toxic profiles of these compounds in vivo were evaluated. Considering the comparable antitumor and anti-inflammatory effects of **SM** epoxides with **SM** and reference drugs (dacarbazine, dexamethasone), **αO-SM** and **βO-SM** can be considered novel promising antitumor and anti-inflammatory drug candidates.

## 1. Introduction

The modification of the natural compounds with various functional groups is one of the main strategies used in drug discovery and development. Epoxide (oxirane, oxacyclopropane)—cyclic ether with an oxygen atom in a saturated three-membered ring—is an important functional group in organic and medicinal chemistry [1,2,3]. Despite its structural simplicity, epoxide is a very versatile moiety with a suitable synthetic balance between stability and reactivity, often used as a synthetic intermediate and valuable pharmacophoric agent. Atoms in the epoxide ring are bonded to each other through σ bonds, and both C-O bonds are polar, given the higher electronegativity of the O atom, which is sp3 hybridized. This is why the oxirane ring can undergo regio- and stereoselective functionalization with various agents (nucleophiles, electrophiles, acids, bases, reducing and oxidizing agents) to produce chiral compounds of diverse structural patterns.

On the one hand, epoxides have been identified for many years as biologically hazardous compounds due to their ability to bind covalently to the nucleophilic centers of proteins and nucleic acids [4,5]. On the other hand, due to their high reactivity, epoxides are used as alkylating or cross-linking agents, indicating their high biological potential as anticancer agents [3]. Mechanistic studies have shown that apoptosis seems to be the most common type of cell death induced by the epoxy-containing derivatives [6,7,8]; however, such compounds can also induce necrosis [3], and thus comprehensive evaluation of the molecular mechanisms underlying their cytotoxicity is required. Despite controversy over the definition of biologically hazardous functional groups, epoxides are of considerable interest in medicinal chemistry, not only as anticancer candidates but also as important agents for treating other pathologies. Thus, an interesting class of epoxy compounds is semisynthetic triterpenoids. In addition to their proven anticancer effect [9,10], they show an agonistic activity against FXR [11,12], a receptor with an emerging role in regulating liver inflammation, anti-hepatitis virus B potency [13] and antithrombotic activity [14].

Previously, our research group reported that a semisynthetic derivative of 18βH-glycyrrhetinic acid (18βH-GA), soloxolone methyl (methyl-2-cyano-3,12-dioxo-18βH-olean-9(11),1(2)-dien-30-oate; **SM**), obtained by direct modification of the A- and C-rings of 18βH-GA, effectively inhibits tumor cell growth in vitro [15] and in vivo [16], impedes their epithelial-mesenchymal transition [17], and significantly suppresses B16 melanoma cell metastasis in a murine model [17]. Along with pronounced antitumor potential, **SM** displays anti-inflammatory activity, inhibiting macrophage functionality in vitro [18] and inflammatory processes in a range of murine models, including phlogogen-induced paw edema [16] and peritonitis [18], influenza A-induced pneumonia [19] and lipopolysaccharide-driven acute lung injury [20].

The structure–activity relationship analysis indicates that the high bioactivity of cyanenone triterpenoids, including **SM**, is associated with the reversible addition of thiol groups of proteins to C-1 due to the dual activation of this olefin by keto and cyano groups [21]. However, the reversible nature of the interaction between cyanenone triterpenoids and thiols hinders attempts to identify pharmacologically relevant targets of these compounds and characterize their action more comprehensively. Recently, 1α,2α-epoxide of bardoxolone methyl was synthesized as a cyanenone analog, irreversibly reacting with model thiols [22]. Moreover, Nakagami et al. [23] and Inoue et al. [24] reported the evaluation of a novel α-epoxy derivative of bardoxolone methyl RS-9 obtained by biotransformation of the triterpenoid with *Chaetomium globosum* SANK 10312. Despite the proven high bioactivity of these compounds, the studies of their pharmacological potential are limited to the evaluation of their stimulating effect on Nrf2 signaling [22,23,24], protective activity against light- and oxygen-induced cellular damage in vitro and retinopathy in vivo [23,24] and metabolic biotransformations in vitro [25]. Wong et al. [22] and Nakagami et al. [23] studied the cytotoxicity of α-epoxy derivatives of bardoxolone methyl in a range of mammalian cell cultures; however, they did not identify by which mechanisms these triterpenoids induce cell death. Moreover, it remains unexplored how markedly the replacement of double-bound within the cyanenone pharmacophore group of bardoxolone methyl and its analogs with epoxide moiety modulates their cytotoxic profile as well as antitumor and anti-inflammatory potential in vivo. Furthermore, to the best of our knowledge, the evaluation of pharmacological properties of only bardoxolone methyl α-epoxide was reported, whereas the bioactivity of its β-epimer has not yet been studied.

Herein, we report the synthesis and evaluation of the anticancer and anti-inflammatory potential of 1α,2α- (**αO-SM**) and 1β,2β- (**βO-SM**) epoxides of **SM**. The screening of cytotoxicity of the novel **SM** epoxides in tumor cells and investigation of their pro-apoptotic activity were performed. Thereafter, the effects of **αO-SM** and **βO-SM** on doxorubicin uptake by murine B16 melanoma cells and the pro-metastatic phenotype of B16 cells were explored. Revealed anti-metastatic activity of the investigated **SM** epoxides was further verified in a murine model of B16 melanoma, followed by their toxic profiling in vivo. Finally, the anti-inflammatory potency of **αO-SM** and **βO-SM** was explored in murine J774 macrophages in vitro and carrageenan-induced peritonitis in vivo. The comparison of the bioactivities of **αO-SM** and **βO-SM** with **SM** allows us to make fundamental conclusions about the influence of the replacement of a double bond in the **SM** structure with an epoxy group, as well as the configuration of the epoxy ring on the antitumor and anti-inflammatory potential of these triterpenoids.

## 2. Results and Discussion

### 2.1. Chemistry

**SM** was used as the starting material for the synthesis (Figure 1).

The α-epoxide (**αO-SM**) was synthesized from **SM** by direct oxidation with aqueous hydrogen peroxide (30%) in acetonitrile. Subsequent purification by column chromatography gave (**αO-SM**) in 70% yield (Figure 1).

In order to obtain the β-epoxide isomer (**βO-SM**), we used an alternative approach to the formation of the epoxy group by the elimination of halohydrines (Figure 1). The intermediates, bromohydrines, were obtained from the interaction of **SM** with bromosuccinimide (NBS) in the acetonitrile–water (3:1, *v*/*v*). ^13^C NMR analysis of the reaction mixture revealed two C1-hydroxy derivatives without traces of C-2-hydroxy ones: *δ* = 77.38 (d, C-1) (major isomer) and 82.12 (d, C-1) (minor isomer) in a ratio of 4.3:1. An attempt to form an epoxy cycle using an alkaline alcohol solution resulted in a mixture of unidentifiable compounds. At the same time, we found that adding calcium carbonate to a mixture of bromohydrins in the acetonitrile–water (3:1, *v*/*v*) leads to the formation of epoxides **βO-SM** and **αO-SM** presented in the reaction mixture in a ratio of 1:3. Epoxides **βO-SM** and **αO-SM** were isolated individually by column chromatography in 22% and 54% yield, respectively. It should be noted that, contrary to our expectations, β-epoxide was a minor product of this reaction, as well as the fact that the ratio of epoxides is different from the ratio of bromohydrins. Therefore, this process will be further investigated.

To determine the steric structure of the **βO-SM** and **αO-SM** epoxides, we made a comparative analysis of the ^1^H-^1^H NOESY spectra (Appendix A). The correlations were observed between the βH-1, H-11 and CH_3_-25 for **αO-SM**. While for **βO-SM,** the correlations were observed between αH-1 and H-11 and between H-11 and CH_3_-25, but there was no correlation between βH-1- and CH_3_-25. The correlation was also observed between αH-1 and αH-5 in the ^1^H-^1^H NOESY spectra of **βO-SM.**

### 2.2. Evaluation of Antitumor Potential of Novel Epoxides of SM

#### 2.2.1. Cytotoxicity

In the first step, the cytotoxicity of **αO-SM** and **βO-SM** with respect to a panel of tumor cells, including human hepatocellular carcinoma HepG2 cells, duodenal carcinoma HuTu-80 cells, cervical carcinomas KB-3-1 and KB-8-5 (with multidrug-resistant phenotype), neuroblastoma KELLY and non-small cell lung carcinoma A549 cells, as well as murine hepatoma Hepa1-6 cells and melanoma B16 cells was studied. Non-transformed human hFF3 fibroblasts were used as non-malignant control cells. The cells were treated with **αO-SM** and **βO-SM** for 24 h, and cell viability was measured by MTT assay. In these experiments, **SM**, a parent compound for **αO-SM** and **βO-SM,** was used as a positive control. The obtained IC_50_ values of the studied compounds are listed in Table 1.

As shown in Table 1, SM epoxides exhibited marked cytotoxicity against both tumor and non-malignant cells in low micromolar concentrations (IC_50_**^αO-SM^** = 0.7–5.2 µM; IC_50_**^βO-SM^** = 0.7–2.1 µM). The presence of an epoxy moiety instead of a double bond in the ring A of **SM** was found to slightly decrease its overall cytotoxicity (average IC_50_**^α/βO-SM^** = 2.0 µM versus IC_50_**^SM^** = 0.8 µM). Notably, **αO-SM** was less toxic for tumor cells compared with its β-epoxide counterpart (average IC_50_**^αO-SM^** = 2.1 µM versus IC_50_**^βO-SM^** = 1.6 µM). Despite this, **αO-SM** showed the highest selectivity toward malignant cells among all evaluated triterpenoids (maximal selectivity index (SI) = 7.4 (KB-8-5)), whereas **βO-SM** was less selective than **SM** (average SI**^βO-SM^** = 1.2 versus SI**^SM^** = 2.1) (Table 2). The obtained results also revealed that neuroblastoma KELLY cells were more susceptible to the evaluated compounds compared with other tumor cell lines (average IC_50_^KELLY^ = 0.7 µM versus IC_50_^other tumor cells^ = 1.6 µM) (Table 1). Interestingly, the cytotoxicity of SM epoxides and SM itself did not depend on the MDR phenotype of tumor cells: IC_50_ values of the compounds in both KB-3-1 and KB-8-5 cells (MDR phenotype) [26] are similar (Table 1). Considering comparable susceptibility of the studied tumor cells to SM epoxides, murine melanoma B16 cells characterized by invasive phenotype and known tumorigenicity in mice were selected as model cells for further mechanistic studies.

Thus, the performed cytotoxic profiling of novel derivatives of **SM** clearly showed that the formation of the epoxide ring in the cyanoenone pharmacophore group did not abrogate the high toxicity of the triterpenoid in tumor cells and only slightly decreased this parameter (overall cytotoxicity of triterpenoids decreased in the order: **SM** > **βO-SM** > **αO-SM**). Furthermore, the formation of an epoxy moiety instead of a double bond in a cyanoenone-bearing triterpenoid scaffold can be considered a novel approach to increasing the selectivity of **SM** against tumor cells; however, this hypothesis needs further confirmation in animal studies.

#### 2.2.2. Novel SM Epoxides Trigger Mitochondrial Apoptosis in B16 Melanoma Cells

Given the proven ability of **SM** and its analogs to stimulate apoptosis in tumor cells [15,27] and the known ability of some epoxide-bearing derivatives of triterpenoids to induce necrosis [3], we next sought to determine which type of cell death is induced by novel SM epoxides in tumor cells. Double staining of compounds-treated B16 melanoma cells with Annexin V-FITC and propidium iodide (PI) revealed that both **αO-SM** and **βO-SM** induced apoptosis in a dose-dependent manner with a similar efficiency leading to the accumulation of around 74% of apoptotic cells after incubation with each epoxide at 1 µM for 24 h (Figure 1A). Under these conditions, **SM** was found to be more active compared with its epoxides: **SM** at 0.5 µM induced apoptosis in 44% of melanoma cells, whereas **αO-SM** and **βO-SM** (0.5 µM) induced apoptotic changes in 11.7% and 16.1% of cells, respectively (Figure 1A). The treatment of B16 cells with all evaluated triterpenoids did not cause massive cell necrosis: the content of necrotic cells in the experimental samples did not exceed 10% (Figure 1A, left upper quadrants).

Further analysis of the mitochondrial membrane potential (Δψ_M_) in B16 melanoma cells demonstrated that **αO-SM** and **βO-SM** caused an accumulation of the cells with depolarized mitochondria in a dose-dependent manner, increasing their number by 1.7 and 2.6 times at 1 µM, respectively, compared with the control (Figure 1B). Similar to the previous test, **SM** was found to exhibit more pronounced bioactivity than its epoxides, increasing the number of cells with reduced Δψ_M_ 3.5-fold already at 0.5 µM compared with the control (Figure 1B). Quantitative analysis of the ratio of aggregates (high Δψ_M_) to monomers (low Δψ_M_) of JC-1, a Δψ_M_-sensitive mitochondrial probe, revealed statistically significant differences in comparison with the control only for B16 cells incubated with the highest epoxides’ concentration (1 µM) (Figure 1C), and **βO-SM** was more active than **αO-SM**. These findings suggest that epoxides of **SM** can trigger apoptosis in tumor cells via a mitochondrial-dependent mechanism.

Next, a double-check revealed the apoptosis-inducing effect of the novel **SM** epoxides, and the activity of executioner caspase-3/-7 in B16 cells was explored. As expected, both **αO-SM** and **βO-SM** at 1 µM increased caspase-3/-7 activity in the cells with similar potency by approximately 2-fold compared with the control; SM at the same concentration was found to have a more pronounced effect on this apoptosis-related parameter (Figure 1D).

In summary, these findings suggest that the replacement of a double bond with an epoxy moiety within the cyanoenone pharmacophore group of **SM** only slightly reduced its pro-apoptotic activity: **αO-SM** and **βO-SM** induced the mitochondrial-dependent apoptosis of B16 melanoma cells, and **βO-SM** more effectively reduced Δψ_M_ in tumor cells compared with its α-counterpart. The obtained results are in line with the published data: previously, we demonstrated the ability of **SM** to induce the mitochondrial pathway of apoptosis in human KB-3-1 cervical carcinoma cells [15].

#### 2.2.3. SM and SM Epoxides Increased Intracellular Accumulation of Doxorubicin in B16 Melanoma Cells Probably via Direct Interaction with Xenobiotic Efflux Pumps

Due to the proven multitarget profile of pentacyclic triterpenoids [28], these compounds not only induce the death of tumor cells by themselves but also effectively potentiate the cytotoxicity of known antitumor drugs [29], which attracted increasing interest of researchers to pentacyclic triterpenoids as promising components for combination antitumor chemotherapy. In light of this, we questioned whether SM and **SM** epoxides influence the cytotoxicity of doxorubicin in B16 melanoma cells. To address this, B16 cells were incubated with the compounds and doxorubicin for 72 h, followed by the measurement of cell viability using an MTT assay. As shown in Figure 2A, the combined action of all tested triterpenoids with doxorubicin significantly increased the cytotoxicity of the latter. We hypothesized that the observed effect could be explained by the ability of SM and SM epoxides to modulate doxorubicin uptake in B16 cells since some pentacyclic triterpenoids were found to increase the intracellular accumulation of doxorubicin in tumor cells [30,31]. To test this hypothesis, flow cytometry analysis of B16 cells treated with investigated compounds and doxorubicin was performed. As shown in Figure 2B,C, all triterpenoids indeed reinforced doxorubicin uptake in B16 cells, and **αO-SM** demonstrated potency somewhat higher than **SM**, whereas **βO-SM** had a lower effect compared with its α-counterpart and **SM**.

Considering the key role of ATP-binding cassette (ABC) transporters in the efflux of xenobiotic molecules from tumor cells [32], our attention further turned to the study of the probable direct interaction of **SM** epoxides and **SM** with P-glycoprotein (ABCB1), MRP1 (ABCC1) and MXR1 (ABCG2), ABC transporters, which play important functions in the regulation of drug resistance and high malignancy of melanoma [33,34,35]. Performed molecular docking studies demonstrated that all evaluated triterpenoids could properly fit into the transmembrane domain of the listed transporters with low binding energies, forming hydrogen bonds with key amino acid residues crucial for the pump activity of these proteins (Figure 2D):In the case of P-glycoprotein, **SM** and **αO-SM** formed a hydrogen bond with Tyr307, which is involved in regulating the ATP hydrolytic activity of the protein [36] and is a major interaction partner of the novel benzophenone sulfonamide-based inhibitor of P-glycoprotein [37]. The binding pockets of **SM** and **SM** epoxides were also found to involve a hydrogen bond with Gln347, which is a well-known inhibitor-binding residue in P-glycoprotein [38,39,40].In the case of MRP1, **αO-SM** formed a hydrogen bond with Arg1248, playing an important function in the positioning of substrates within the drug-binding pocket of the protein [41,42] and involving the amino acid interaction network of the novel flavonoid-based inhibitor of MRP1 [43]. The binding sites of **SM** and **βO-SM** were found to involve a hydrogen bond with Ser605, controlling the substrate specificity of MRP1 [44].Our docking studies indicated that the binding of SM and its epoxides to MXR1 involved hydrogen bonds with Thr402 and Val442, a key structural determinant responsive to interhelical interaction and, thus, the overall transport activity of the protein [45,46] and known drug-binding residue [47,48,49], respectively.

Interestingly, the performed computing of the binding energies of investigated triterpenoids to ABC transporters revealed their high correlation with the level of doxorubicin uptake in triterpenoid-treated B16 cells: the efficiency of the binding of **SM** epoxides and **SM** to the mentioned efflux transporters decreased in the order **αO-SM** ≥ **SM** > **βO-SM**, as in the case of doxorubicin uptake assay (compare Figure 2C,D). The obtained results agree with the published data; recently, Rybalkina et al. demonstrated the ability of semisynthetic pentacyclic triterpenoids with modified ring A to inhibit the efflux activity of P-glycoprotein in multidrug resistant HBL-100/Dox cells [50].

Taken together, our findings demonstrated that **SM** and **α/βO-SM** could probably increase the intracellular accumulation of doxorubicin via the inhibition of the pump activity of key efflux transporters, such as P-glycoprotein, MRP1 and MXR1. We should note here that this study aimed to estimate the combined cytotoxic profile of SM and its epoxides with doxorubicin and perform initial mechanistic insights into this effect only. The full-scale investigation of synergistic/additive effects of **SM** and **α/βO-SM** with doxorubicin in tumor cells and evaluation of their influence on the P-glycoprotein efflux activity in multidrug resistant malignant cells are the objectives of our next study.

#### 2.2.4. αO-SM and βO-SM Effectively Suppress Metastasis-Related Characteristics of B16 Melanoma Cells In Vitro

It is known that pentacyclic triterpenoids, including SM and its analogs, not only induce the death of tumor cells but also significantly suppress their pro-metastatic phenotype [17,51]. To explore this possibility, the effect of novel epoxides of **SM** on the clonogenicity, motility and adhesiveness of B16 melanoma cells was evaluated. To avoid the influence of toxic effects of evaluated compounds on the listed parameters, their submicromolar concentrations were used. As shown in Figure 3A,B, novel **SM** epoxides significantly reduced the clonogenic potency of B16 cells in a dose-dependent manner, and **βO-SM** was found to have a more promising effect than its α-counterpart, suppressing the colony formation rate of B16 cells at 0.5 µM by 1.9-times compared with the control versus 1.2-fold reduction revealed for **αO-SM** at the same concentration. As shown previously, SM was markedly more bioactive than its epoxides, leading to an average 4.1-fold suppression of clonogenicity of melanoma cells compared with the control at all used concentrations (Figure 3A,B).

The performed scratch assay demonstrated that the triterpenoids significantly inhibited the migration potential of B16 cells with different efficiencies, and their suppressive effect on tumor cell motility decreased in the order **SM** > **βO-SM** > **αO-SM** (Figure 3C,D). It was found that **αO-SM** and **βO-SM** at 1 µM reduced the wound closure rate of B16 cells 1.7- and 2.3-fold compared with the control, respectively, whereas **SM** already at 0.5 µM declined this parameter 2-fold (Figure 3D).

Along with the obtained results, the cell adhesion assay also confirmed the high potency of novel epoxides of **SM** as promising anti-metastatic candidates. It was shown that the incubation of B16 cells with SM epoxides for 24 h significantly inhibited the adhesiveness of tumor cells to tissue-culture-treated polystyrene wells (Figure 3E). Notably, neither the formation of the epoxy group in **SM** structure nor the stereo-positions of epoxide moiety in **α/βO-SM** did not influence the anti-adhesive potential of the triterpenoids: all studied compounds were found to exhibit comparable effects on B16 cell adhesiveness (Figure 3E).

In summary, these data indicate that epoxides of **SM** not only have a direct pro-apoptogenic effect on B16 melanoma cells but also effectively inhibit their metastasis-related properties, which confirms the expediency of further, more detailed study of the anti-metastatic activity of **α/βO-SM** in an animal model. As in the case of cytotoxic profiling and evaluation of Δψ_M_, the overall anti-metastatic potential of evaluated triterpenoids in vitro decreased in the order **SM** > **βO-SM** > **αO-SM**.

#### 2.2.5. αO-SM and βO-SM Demonstrate Anti-Metastatic Activity on the B16 Melanoma Model In Vivo

Previously, we reported that **SM** effectively inhibits the metastasis of B16 melanoma cells in a murine model [17]. To examine how strongly the replacement of the double bond in the cyanoenone pharmacophore group of **SM** with the epoxide moieties affects this activity, the anti-metastatic potency of **αO-SM** and **βO-SM** was studied using an in vivo model of B16 melanoma transplanted intravenously (i.v.) and formed lung metastases (Figure 4). Tumor-bearing mice were administered intraperitoneally (i.p.) with **αO-SM** and **βO-SM** at a dosage of 30 mg/kg three times a week starting on day 3 after tumor transplantation (Figure 4A). **SM** and the cytotoxic drug dacarbazine (**DTIC**) successfully applied in clinical practice against metastatic melanoma [52,53] were used as reference drugs. Mice receiving vehicles (10% Tween-80, i.p.) were considered a control group. Eight injections of the compounds were made in total. The mice were sacrificed on day 21 after tumor transplantation. The number of surface metastases and metastasis inhibition index (MII) (see Section 4) in the lungs were estimated. Mice body weight was measured during the experiment; organ indices and morphological changes in the liver and kidneys were evaluated to assess the toxicity of the studied triterpenoids.

Obtained results demonstrated that the administration of **αO-SM** and **βO-SM** led to a 6- and 4.4-fold decrease in the number of surface metastases in the lungs of B16 melanoma-bearing mice compared with the vehicle-treated group, respectively (Figure 4B), whereas **SM** and **DTIC** exhibited lower efficiency, reducing the number of surface lung metastases 3.7- and 2.3-fold compared with vehicle, respectively (Figure 4B). The calculation of MII confirmed the high anti-metastatic potential of novel epoxides of **SM**: it was found that MII values for **αO-SM**- and **βO-SM**-treated groups were 83.3 ± 2.5% and 77.2 ± 4.8%, respectively; **SM** and **DTIC** demonstrated lower MII rate (73.3 ± 10.1% and 56.9 ± 16.1%, respectively) (Figure 4C).

Thus, the obtained data clearly indicate the pronounced anti-metastatic potency of **αO-SM** and **βO-SM**, which was higher than that of the parent compound **SM** and **DTIC** with well-proven anti-metastatic activity.

#### 2.2.6. αO-SM and βO-SM Have Toxic Effects in B16 Melanoma-Bearing Mice

Despite the observed pronounced anti-metastatic effect of **αO-SM** and **βO-SM** on B16 melanoma, their injections in tumor-bearing mice were found to have general toxic effects. Analysis of mouse body weight demonstrated that injections of novel epoxides of **SM** did not cause its loss until day 17 after tumor transplantation; however, at the end of the experiment marked decline in this parameter was identified (Figure 4D). Evaluation of the body weight of mice before their sacrifice on day 21 revealed that the observed changes in **αO-SM**- and **βO-SM**-treated groups were statistically significant: administration of these triterpenoids led to weight loss of 9.4% and 7.2%, respectively, compared with the vehicle-treated group (Figure 4E). Further evaluation of organ indices demonstrated a tendency of **αO-SM** and **βO-SM** toward hepato- and nephrotoxicity: these compounds were found to increase in hepatic indices by 1.3- and 1.2-times and renal indices by 1.7- and 1.4-times, respectively, compared with vehicle (Figure 4F). Additionally, a 1.3-fold increase in the hepatic index in **SM**-treated mice was revealed compared with vehicle-treated ones (Figure 4F). Evaluation of lung, heart and spleen indices did not identify any toxic effects of the investigated compounds on these organs.

For a more detailed assessment of liver and renal toxicity, histological and morphometric analyses of livers and kidneys sampled from healthy mice and vehicle-, **SM**-, **αO-SM**- and **βO-SM**-treated tumor-bearing mice were performed. The volume densities of destructive changes in the liver are represented by dystrophies (reversible changes), and necrosis (irreversible changes), as well as the numerical densities of binuclear hepatocytes reflecting the regenerative potential of the liver, were evaluated. The development of B16 melanoma in vehicle-treated mice was found to have distinguishable toxic effects on the liver manifested by an increase in the total destructive changes up to 24% from the entire liver parenchyma, predominantly due to necrosis (Table 3, Figure 4G). The administration of **SM** and **βO-SM** in tumor-bearing mice had no additional adverse effects on the liver compared with the vehicle group, whereas treatment with **αO-SM** led to a 1.9-fold and 1.7-fold increase in the volume densities of dystrophies and necrosis in liver tissue, respectively, compared with the vehicle-treated group (Table 3, Figure 4G). **DTIC** was found to increase dystrophies in the liver parenchyma only and not induce necrosis. Analysis of hepatic recovery characteristics demonstrated that **αO-SM** significantly suppressed the regenerative potential of the liver, decreasing the numerical density of binuclear hepatocytes by 2.2- and 1.9-fold compared with healthy mice and vehicle-treated tumor-bearing animals, respectively. Interestingly, other triterpenoids and **DTIC** did not affect this parameter (Table 3; Figure 4G).

Morphometric analysis of the kidneys, including evaluation of the number of normal epitheliocytes of proximal tubules and epitheliocytes with dystrophic and necrotic changes, revealed that the development of B16 melanoma in vehicle-treated mice increased the number of dystrophic and necrotic epitheliocytes that reduced normal kidney tissue to approximately 77% compared with the healthy group (Table 4, Figure 4G). Administration of **αO-SM** was found to cause a 1.5-fold enhancement of the destructive changes in kidneys due to both dystrophy and necrosis of the epithelium of proximal tubules compared with vehicle-treated animals, whereas injections of its β-counterpart induced a statistically insignificant increase in dystrophic changes in the renal parenchyma being potentially reversible (Table 4, Figure 4G). Administration of **SM** and **DTIC** did not stimulate destructive processes in kidney tissue: the number of dystrophic and necrotic epitheliocytes in the kidney of **SM**- and **DTIC**-treated mice was comparable with that of the vehicle-treated group (Table 2, Figure 4G).

In summary, the obtained results demonstrate that the replacement of the double bond in the pharmacophore group of **SM** with an epoxide ring does not generally influence the anti-metastatic potency of the triterpenoids in vivo; however, it does lead to a tendency to increase its antitumor effect, and imparts some toxic effects on liver and kidneys, which may be associated with the usage of a non-optimized treatment regimen. The observed general toxicity of **αO-SM** and **βO-SM** can be explained by the fact that epoxide-bearing compounds can form irreversible covalent bonds with cysteine residues of proteins [22] which leads to their accumulation in key metabolizing (liver) and excretive (kidney) organs. Nevertheless, **βO-SM** exhibiting equilibrium between antitumor activity and hepato-/nephrotoxicity might be considered a promising anti-metastatic candidate, assuming the careful selection of used dosages and treatment regimen. Additionally, it is worth mentioning that optimizing the therapeutic scheme and dosage could significantly reduce general and organ toxicity without the loss of the anti-metastatic activity of **SM** epoxides.

### 2.3. Evaluation of Anti-Inflammatory Potential of Novel Epoxides of SM

#### 2.3.1. αO-SM and βO-SM Exhibit Anti-Inflammatory Potency in J774 Macrophages In Vitro

Given the fact that **SM** and its derivatives not only show high antitumor but also promising anti-inflammatory effects proven on various inflammatory murine models [16,18,19,20,54], we next questioned whether the formation of epoxide moiety in the pharmacophore group of SM affects the anti-inflammatory potential of the triterpenoid. To explore this, the effects of **αO-SM** and **βO-SM** on (i) the ability of murine J774 macrophages to produce nitric oxide (II) (NO), a known pro-inflammatory mediator [55], and (ii) the migration capacity of macrophages, playing an important role in the inflammatory process [56], were evaluated.

In the first step, the cytotoxicity of **αO-SM**, **βO-SM** and **SM** in J774 cells was assessed to identify their non-toxic concentrations required for the experiments with macrophages. As shown in Figure 5A, **SM** epoxides and **SM** did not affect the viability of the cells in the concentration range up to 1 µM and 2 µM, respectively. Further, the Griess colorimetric assay demonstrated that **βO-SM** at 0.5 µM 1.5-fold reduced the synthesis of NO by IFNγ/LPS-stimulated J774 macrophages compared with the control, whereas its α-counterpart did not display any inhibitory activity at used concentrations at all (Figure 5B). SM was found to have a more pronounced effect than its epoxides: an incubation of IFNγ/LPS-stimulated J774 cells with SM at 0.5 µM resulted in a 2.7-fold reduction of NO production compared with the control (Figure 5B).

In addition to NO measurement, the performed scratch assay also confirmed the inhibitory effect of **SM** and **SM** epoxides on macrophage functionality. It was found that the incubation of J774 cells with these compounds significantly suppressed macrophage motility (Figure 5C). We showed that **βO-SM** at 2 µM more effectively reduced the migration capacity of J774 cells compared with its α-epimer, decreasing this parameter by 40.4% versus 12.7% for **αO-SM**-treated cells (Figure 5D). **SM** exhibited the highest inhibitory effect in this assay, leading to the suppression of macrophage motility by 32.9% at 1 µM (Figure 5D).

Taken together, the obtained results clearly demonstrate that the formation of epoxide moiety within ring A of SM does not abrogate its anti-inflammatory activity in vitro and only somewhat decreases the efficiency of the latter. As in the case of antitumor assays, the anti-inflammatory potency of the investigated triterpenoids decreased in the order of **SM** > **βO-SM** > **αO-SM**.

#### 2.3.2. αO-SM and βO-SM Effectively Inhibit Carrageen-Induced Peritonitis In Vivo

The anti-inflammatory activity of **αO-SM** and **βO-SM** revealed in vitro was further verified in a murine model of carrageenan-driven peritonitis (Figure 5E). As shown in Figure 5E, the intraperitoneal injection of carrageenan in vehicle-treated mice caused an acute inflammatory reaction in the peritoneal cavity characterized by a 4.7-fold increase in the number of leukocytes (Figure 5F), represented predominantly by the granulocyte subpopulation (Figure 5G), and a 63.9-fold increase in the level of pro-inflammatory cytokine TNFα in the peritoneal exudate compared with healthy animals (Figure 5H). Administration of **SM**, **αO-SM** and **βO-SM** 1 h prior to peritonitis induction led to the significant inhibition of carrageenan-induced inflammation in the peritoneal cavity, manifested by 1.9-, 2.2- and 2.0-fold decreases in the number of total leukocytes and 9.3-, 5.2- and 9.9-fold decreases in the TNFα level, respectively, compared with vehicle-treated animals (Figure 5F,H). Dexamethasone, used as a reference drug, had a similar inhibitory effect on leukocyte infiltration as triterpenoids (Figure 5F); however, it showed a lower influence on TNFα production (Figure 5H). Notably, **SM** was found to restore the leukocyte subpopulation in the inflamed peritoneal cavity to a healthy level (Figure 5H), and it is obvious that the cyanoenone pharmacophore group of **SM** plays a key role in this activity since the formation of an epoxide moiety in this group fully abrogated the observed effect (Figure 5H).

In summary, our findings clearly demonstrate that the epoxide ring formation in ring A of **SM** does not affect its anti-inflammatory properties in vivo; however, it abolishes its blocking effect on carrageenan-stimulated granulocyte infiltration in the peritoneal cavity. Considering the similar levels of anti-inflammatory activity of SM and its epoxides to dexamethasone, investigated triterpenoids can be considered a promising platform for developing novel anti-inflammatory candidates.

## 3. Conclusions

The performed comprehensive evaluation of the pharmacological effects of two novel epoxides of **SM** (**αO-SM** and **βO-SM**) on tumor and immune cells in vitro and in vivo clearly demonstrated that these compounds (i) induced the death of tumor cells at low micromolar concentrations via the induction of mitochondrial pathway of apoptosis, (ii) increased doxorubicin uptake by tumor cells, (iii) effectively inhibited pro-metastatic properties of B16 melanoma cells, (iv) suppressed macrophage functionality in vitro and (v) blocked inflammation in mice with carrageenan-induced peritonitis. Structure–activity relationship analysis revealed that the replacement of the double bond in the pharmacophore group of SM with epoxide moiety resulted in a somewhat reduction of its bioactivity in vitro, and **βO-SM** displayed a more pronounced antitumor and anti-inflammatory potency compared with its α-counterpart. Interestingly, in murine models of B16 melanoma metastasis and carrageenan-driven peritonitis, SM and its epoxides exhibited similar efficiencies between themselves without obvious preference for any epimers. Evaluation of the toxicity-associated parameters in vivo indicated that **αO-SM** and **βO-SM** have somewhat hepato- and nephrotoxicity, and **βO-SM** showed a more safe profile than its α-counterpart. Considering the comparable antitumor and anti-inflammatory effects of **αO-SM** and **βO-SM** in vivo with reference drugs **DTIC** and dexamethasone, respectively, these compounds can be considered novel promising antitumor and anti-inflammatory drug candidates.

## 4. Materials and Methods

### 4.1. Chemistry

#### 4.1.1. General Experimental Procedures and Reagents

Elemental analyses were carried out on an Automatic CHNS-analyser EURO EA3000. Analyses indicated by the symbols of the elements were within ±0.4% of the theoretical values. Melting points were determined on a METTLER TOLEDO FP900 thermosystem and were uncorrected. Optical rotations were measured with a PolAAr 3005 polarimeter. The elemental composition of the products was determined from high-resolution mass spectra recorded on a DFS (double-focusing sector) Thermo Electron.^1^H and ^13^C NMR spectra were measured on Bruker spectrometers: AV-600 (600.30 MHz for ^1^H and 150.95 MHz for ^13^C). The solutions of each compound were prepared in CDCl_3_. Chemical shifts were recorded in *δ* (ppm) using *δ* 7.24 (^1^H NMR) and *δ* 76.90 (^13^C NMR) of CHCl_3_ as internal standards. Chemical shift measurements were given in ppm and the coupling constants (*J*) in hertz (Hz). The structure of the compounds was determined by NMR using standard one-dimensional and two-dimensional procedures (^1^H-^1^H COSY, ^1^H-^1^H NOESY, ^1^H-^13^C HMBC/HSQC). The purity of the final compounds for biological testing was >98%, as determined by HPLC analysis. HPLC analyses were carried out on a MilichromA-02, using a ProntoSIL 120-5-C18 AQ column (BISCHOFF, 2.0 × 75 mm column, grain size 5.0 lm). The mobile phase was Millipore purified water with 0.1% trifluoroacetic acid at a flow rate of 150 µL/min at 35 °C with UV detection at 210, 220, 240, 260 and 280 nm. A typical run time was 25 min with a linear gradient of 0–100% methanol. Flash column chromatography was performed with silica gel (Merck, 60–200 mesh). All courses of all reactions were monitored by TLC analysis using Merck 60 F254 silica gel on aluminium sheets with the eluent CHCl_3_–MeOH (25:1.5 *v*/*v*).

Soloxolone methyl (**SM**) was prepared according to a previously reported method [15]. *N*-Bromosuccinimide (NBS) (99%) was purchased from Sigma Aldrich (St. Louis, MI, USA) and freshly recrystallized from H_2_O.

#### 4.1.2. Oxidation of SM with Hydrogen Peroxide

**SM** (25 mg, 0.05 mmol) was dissolved in acetonitrile (3 mL), then H_2_O_2_ (0.1 mL, 30%, aq) and stirred for 1 h at room temperature. The reaction course was monitored by TLC (CHCl_3_–AcOEt, 25–1.5). Upon reaching full conversion, the reaction mixture was diluted with CH_2_Cl_2_ and Et_2_O, washed sequentially with Na_2_SO_3_ (aq), H_2_O and brine and dried over anhydrous MgSO_4_. The solvent was evaporated to dryness. The crude product (25 mg) was purified by flash column chromatography (silica gel, 0–1% AcOEt in CHCl_3_) to yield **αO-SM** (18 mg, 70%) as a white solid.

Methyl 1α,2α-Epoxy-2β-Cyano-3,12-dioxo-13βH,18βH-olean-9(11)-en-30-oic acid (αO-SM)

Elemental analysis: calculated 73.67% C, 8.31% H, 2.68% N, 15.33% O; found 73.60% C, 8.21% H, 2.62% N. M. p. 252 °C. [decomposition]). [$\alpha_{D}^{25}$] + 108 (*c* 0.20 g/100 mL; CHCl_3_). HRMS: *m*/*z* calcd. for (C_32_H_43_O_5_N)^+^ 521.3136; found 521.3131. ^1^H NMR (CDCl_3_, 600 MHz): *δ* = 6.00 (s, 1H, H-11), **4.31 (s, 1H, H-1)**, 3.72 (s, 3H, O*CH_3_*-31), 3.03 (d, 1H, *J_13β,18β_* = 4.7, H-13β), 2.21 (ddd, 1H, ^2^*J*_19e,19a_ = 13.2, ^3^*J*_19e_,_18_ = 3.2, ^4^*J*_19e21e_ = 2.8, H-19e), 2.02 (ddd, 1H, ^3^*J*_18β_,_19a_ = 13.2, ^3^*J*_18β_,_13β_ = 4.7, ^3^*J*_18β_,_19e_ = 3.2, H-18β), **1.94 (m, 2H, H-5a; H-21e)**, 1.88 (m, H^16a^), 1.80 (m, H^15a^), 1.67–1.79 (m, 3H; 2H^6^, H^7a^), 1.50 (dm, ^2^*J*_7e,7a_ = 13.5 Hz; H^7e^), 1.46 (m, H^22a^), 1.20–1.32 (m, 3H; H^21a^, H^22e^ H^19a^), 1.08 (m, 1H, H-15e), 0.92 (dm, ^2^*J*_16e,16a_ = 13.3 Hz; H-16e), 1.38 (s, 3H; CH_3_-26), **1.25 (s, 3H; CH_3_-25)**, 1.16 (s, 3H; CH_3_-23), 1.10 (s, 6H; CH_3_-29 and CH_3_-24), 1.03 (s, 3H; CH_3_-27), 0.90 (s, 3H; CH_3_-28). ^13^C NMR (CDCl_3_, 150 MHz), 202.25 (s, C-3), 199.44 (s, C-12), 177.15 (s, C-30), 168.37 (s, C-9), 124.97 (d, C-11), 113.41 (s, *C*N), **68.95 (d, C-1)**, **53.07 (s, C-2)**, 51.49 (q, OCH_3_-31), 48.05 (d, C-13), 45.66 (s, C-8), 44.91 (s, C-4), 43.92 (s, C-20), **42.37 (d, C-5)**, 41.99 (s, C-14), 40.67 (s, C-10), 38.15 (t, C-22), 37.74 (d, C-18), 33.63 (t, C-19), 31.88 (s, C-17), 31.29 (t, C-7), 31.11 (t, C-21), 28.47 (q, C-29), 27.84 (q, C-23), 26.94 (q, C-28), 26.16 (t, C-15) 26.02 (t, C-16), 24.11 (q, C-26), 22.70 (q, C-25), 21.77 (q, C-27), 21.18 (q, C-24), 18.13 (t, C-6).

#### 4.1.3. Treating SM with N-bromosuccinimide (NBS) in the Presence Calcium Carbonate

**SM** (200 mg, 0.40 mmol) was dissolved in acetonitrile–H_2_O (3:1 *v*/*v*, 8 mL), and then NBS (80 mg, 0.45 mmol) was added and stirred for 3 h at room temperature in the darkness. Calcium carbonate (200 mg) was added, and the reaction mixture continued to stir for 72 h. The reaction course was monitored by TLC (CHCl_3_–AcOEt, 25–1.5). Upon reaching full conversion, the reaction mixture was filtered, diluted with CH_2_Cl_2_ and Et_2_O, washed sequentially H_2_O and brine and dried over anhydrous MgSO_4_. The solvent was evaporated to dryness (200 mg). Column chromatography (silica gel, 0–1% AcOEt in CHCl_3_) gave **βO-SM** (46 mg, 22%) as a white solid and **αO-SM** (112 mg, 54%) as a white solid.

Methyl 1β,2β-Epoxy-2α-Cyano-3,12-dioxo-13βH,18βH-olean-9(11)-en-30-oic acid (βO-SM)

M. p. 128.8 °C. [decomposition] Elemental analysis: calculated 73.67% C, 8.31% H, 2.68% N, 15.33% O; found 73.47% C, 8.27% H, 2.72% N. [$\alpha_{D}^{25}$] + 78 (*c* 0.20 g/100 mL; CHCl_3_). HRMS: m/z calcd. for (C_32_H_43_O_5_N)^+^ 521.3136; found 521.3131. ^1^H NMR (CDCl_3_, 600 MHz): *δ* = 5.95 (s, 1H, H-11), **4.06 (s, 1H, H-1),** 3.72 (s, 3H, O*CH_3_*-31), 3.07 (d, 1H, *J_13β,18β_* = 4.6, H-13β), 2.19 (ddd, 1H, ^2^*J*_19e,19a_ = 13.2, ^3^*J*_19e_,_18_ = 3.2, ^4^*J*_19e21e_ = 2.8, H-19e), 2.03 (ddd, 1H, ^3^*J*_18β_,_19a_ = 13.2, ^3^*J*_18β_,_13β_ = 4.6, ^3^*J*_18β_,_19e_ = 3.2, H-18β), 1.94 (dm, 1H, ^2^*J*_21e,21a_ = 13.5, H-21e), 1.89 (m, 1H, H-16a), 1.83 (m, 1H, H-15a), **1.79 (dd, 1H, ^3^*J*_5a,6a_ = 10.7, ^3^*J*_5a,6e_ = 3.9, H-5a**), 1.60–1.74 (m, 3H; 2H^6^, H^7a^), 1.50 (dm, ^2^*J*_7e,7a_ = 13.5 Hz; H^7e^), 1.46 (dd, 1H, ^2^*J*_22a,22e_ = 14.0, ^3^*J*_22a,6e_ = 3.9, H-22a), 1.20–1.33 (m, 3H; H^21a^, H^22e^, H^19a^), 1.10 (m, 1H, H-15e), 0.94 (dm, ^2^*J*_16e,16a_ = 13.3 Hz; H-16e), 1.41 (s, 3H; CH_3_-26), 1.26 (s, 3H; CH_3_-23), 1.13 (s, 3H; CH_3_-24), 1.11 (s, 3H; CH_3_-29), **1.10 (s, 3H; CH_3_-25)**, 1.05 (s, 3H; CH_3_-27), 0.91 (s, 3H; CH_3_-28). ^13^C NMR (CDCl_3_, 150 MHz), 200.50 (s, C-3), 199.49 (s, C-12), 177.15 (s, C-30), 171.54 (s, C-9), 127.34 (d, C-11), 114.18 (s, *C*N), **69.17 (d, C-1),** 1.51 (q, OCH_3_-31), **49.61 (d, C-5)**, 48.27 (d, C-13), 48.57 (s, C-4), 45.55 (s, C-8), **45.06 (s, C-2)**, 43.92 (s, C-20), 42.14 (s, C-14), 41.23 (s, C-10), 38.11 (t, C-22), 37.78 (d, C-18), 33.70 (t, C-19), 31.90 (s, C-17), 31.12 (t, C-21), 30.79 (t, C-7), 28.48 (q, C-29), 28.18 (q, C-23), 27.00 (q, C-28), 26.03 (t, C-15), 26.00 (t, C-16), 24.50 (q, C-26), 21.73 (q, C-27), 21.44 (q, C-25), 20.08 (q, C-24), 18.28 (t, C-6).

### 4.2. Biological Evaluations

#### 4.2.1. Cell Lines

Human hepatocellular carcinoma HepG2, human duodenal adenocarcinoma HuTu-80, human cervical carcinoma KB-3-1, human lung adenocarcinoma A549, murine hepatoma Hepa1-6, murine melanoma B16 and murine macrophage-like J774 cells were obtained from the Russian Culture Collection (Institute of Cytology RAS, St. Petersburg, Russia). Human neuroblastoma KELLY cells were purchased from the American Type Culture Collection (ATCC, Manassas, VA, USA). Human cervical carcinoma KB-8-5 cells with the MDR phenotype and human non-transformed hFF3 fibroblasts were kindly provided by Prof. M. Gottesman (National Institutes of Health, Bethesda, MD, USA) and Dr. Olga Koval (Institute of Chemical Biology and Fundamental Medicine, Siberian Branch of the Russian Academy of Sciences (SB RAS), Novosibirsk, Russia), respectively. The cells were cultured in Dulbecco’s modified Eagle’s medium (DMEM) (Sigma Aldrich, St. Louis, MI, USA) (HepG2, HuTu-80, KB-3-1, B16, A549, Hepa1-6, B16, and J774 cells), Roswell Park Memorial Institute (RPMI-1640) medium (Sigma Aldrich, USA), containing 1% (*v*/*v*) GlutaMAX (Gibco, Paisley, UK) (KELLY cells), and Iscove’s Modified Dulbecco’s Medium (IMDM) (Sigma Aldrich, USA) (hFF3 cells). KB-8-5 cells were cultured in DMEM in the additional presence of vinblastine at 300 nM. All culture media contained 10% (*v*/*v*) heat-inactivated fetal bovine serum (FBS) (Gibco, UK) and antibiotic-antimycotic solution, containing penicillin at 10,000 IU/mL, streptomycin at 10,000 µg/mL and amphotericin at 25 µg/mL (MP Biomedicals, Illkirch –Graffenstaden, France). The cells were incubated at 37 °C in a humidified 5% CO_2_-containing air atmosphere (hereafter, standard conditions). **SM** epoxides and **SM** were dissolved in DMSO (stock solution: 10 mM) and stored at −20 °C before the experiment.

#### 4.2.2. Mice

Female C57Bl/6 and Balb/C mice (average weight 20–24 g for both lines) for in vivo studies were obtained from the Vivarium of the Institute of Chemical Biology and Fundamental Medicine SB RAS (Novosibirsk, Russia). The mice were kept in plastic cages (10 animals per cage) under normal daylight conditions. Water and food were provided ad libitum. Experiments were carried out in accordance with the European Communities Council Directive 86/609/CEE. The experimental protocols were approved by the Committee on the Ethics of Animal Experiments at the Institute of Cytology and Genetics SB RAS (Novosibirsk, Russia) (protocol No. 56 from 10.08.2019).

#### 4.2.3. Evaluation of Cytotoxicity of Novel Compounds by MTT Assay

To determine the cytotoxicity of **SM** and **SM** epoxides in the cell lines mentioned above, an MTT test was performed. The cells were seeded in 96-well plates in quadruplicates at 10^4^ cells/well (HepG2, Hepa-1-6, HuTu-80, B16, KB-3-1, KB-8-5, KELLY, A549, hFF3) or 10^5^ cells/well (J774). The plates were incubated under standard conditions for 24 h followed by the replacement of the medium with a fresh medium containing diluted triterpenoids at various concentrations (0.5–5 μM). Cells were incubated in the presence of compounds for 24 h under standard conditions. Thereafter, aliquots of [3-(4,5-dimethylthiazol-2-yl)-2,5-diphenyltetrazolium bromide] (MTT) solution (10 μL, 5 mg/mL) were added to each well and the cells were incubated for an additional 2 h. Then, the medium was removed from the wells, and 100 μL of dimethyl sulfoxide (DMSO) was added to dissolve blue formazan crystals formed within live cells; the optical density was measured at test and reference wavelengths (570 nm and 620 nm, respectively) using a Multiscan RC plate reader (Thermo LabSystems, Helsinki, Finland). The half-inhibitory concentrations (IC_50_) were determined as the compound concentration required to decrease the cell viability to 50% of the control value and were calculated by interpolation from the dose–response curves. The selectivity index (SI) of the compound was calculated as the ratio of its IC_50_ values in non-transformed hFF3 fibroblasts to the IC_50_ values in corresponding malignant cells.

#### 4.2.4. Apoptosis Assay

B16 cells were seeded in 24-well plates at 10^5^ cells/well. The plates were incubated under standard conditions for 24 h. After that, the medium was replaced with a fresh medium containing diluted compounds (**SM** and **SM** epoxides) at 0.5 and 1 μM. Cells were incubated in the presence of compounds for 6, 18 and 24 h under standard conditions. After incubation, the cells were harvested by trypsinization (TrypLE Express (Gibco, Grand Island, NY, USA)) and centrifugation at 400× *g* for 5 min, washed with PBS and resuspended in binding buffer containing 5 µL/well of Annexin-V conjugated with fluorescein-5-isothiocyanate (FITC) and 10 µL/well of propidium iodide (PI) (Annexin V-FITC Apoptosis Detection Kit, Millipore, Bedford, MA, USA). The cells were incubated in darkness at room temperature for 15 min. Finally, the samples were analyzed using a NovoCyte Flow Cytometer (ACEA Biosciences Inc., San Diego, CA, USA). For each sample, 10,000 events were acquired.

#### 4.2.5. Measurement of Mitochondrial Membrane Potential

B16 cells were seeded in 24-well plates in triplicate at 10^5^ cells/well and incubated for 24 h under standard conditions. Thereafter, the medium was replaced with a fresh one without (control) or with the investigated compounds (**SM** epoxides at 0.5 and 1 µM; **SM** at 0.5 µM). The cells were incubated for 18 h, followed by the addition of JC-1 dye solution (5 µL) (Molecular Probes, Invitrogen, Carlsbad, CA, USA) to experimental and control samples and incubation of the cells under standard conditions for an additional 30 min. Then, the cells were washed twice with PBS, detached using TrypLE Express and resuspended in a fresh medium. The samples were analyzed using a NovoCyte Flow Cytometer. For each sample, 10,000 events were acquired.

#### 4.2.6. Evaluation of Caspase-3/-7 Activity

The activity of caspase-3/-7 in B16 cells was evaluated using the Caspase-Glo ^®^ 3/7 assay kit (Promega, Madison, WI, USA) according to the manufacturer’s instructions. Briefly, B16 cells were seeded in a white-walled 96-well plate in triplicate at 10^4^ cells/well and incubated for 24 h under standard conditions. Later, the cells were treated with **SM** and **SM** epoxides at 1 µM for 24 h. After incubation, 100 µL of Caspase-Glo^®^ 3/7 Reagent was added to each well. The plate was incubated for 30 min in darkness at room temperature. The luminescence value was measured using a luminometer CLARIOstar plate reader (BMG Labtech, Ortenberg, Germany).

#### 4.2.7. Cytotoxicity of SM Epoxides in Combination with Doxorubicin

B16 cells were seeded in a 96-well plate in triplicate at 10^4^ cells/well and incubated for 24 h. After that, the cells were treated with triterpenoids (**SM** epoxides at 1 µM; **SM** at 0.5 µM) and doxorubicin (1 µM) separately and in combination for an additional 72 h. Thereafter, the cell viability was measured by the MTT assay described above.

#### 4.2.8. Doxorubicin Accumulation Assay

B16 cells were seeded in a 24-well plate in triplicate at 10^5^ cells/well and incubated under standard conditions for 24 h. Then, the cells were treated with triterpenoids (**SM** epoxides at 1 µM; **SM** at 0.5 µM) and doxorubicin (1 µM) separately and in combination for 3 h, washed twice with PBS and harvested by trypsinization (TrypLE Express (Gibco, Grand Island, NY, USA)) and centrifugation at 400× *g* for 5 min. Thereafter, the cells were resuspended in a fresh culture medium and analyzed by flow cytometry (doxorubicin fluorescence; NovoCyte Flow Cytometer). For each sample, 10,000 events were acquired.

#### 4.2.9. Molecular Docking

Molecular docking of **αO-SM**, **βO-SM** and **SM** with ABC transporters was performed using Autodock Vina [57]. The three-dimensional structures of P-glycoprotein (Protein Data Bank (PDB) ID: 7A6F), MRP1 (PDB ID: 5UJA) and MXR1 (PDB ID: 6FFC) were uploaded from the Research Collaboratory for Structural Bioinformatics (RCSB) Protein Data Bank (https://www.rcsb.org/) (accessed: 4 April 2022) followed by the extraction of co-crystalized ligands and water molecules from the PDB files of the proteins and the addition of polar hydrogen and Gasteiger charges into the protein structures using AutoDock Tools 1.5.7. The two-dimensional structures of **SM** and **SM** epoxides were converted to three-dimensional forms and their geometry was optimized with the MMFF94 force field using Marvin Sketch 5.12 and Avogadro 1.2.0, respectively. All torsions were allowed to rotate freely during molecular docking. The docking parameters were set as follows: P-gycoprotein, center_x = 161.718, center_y = 158.298, center_z = 159.204, size_x = 14, size_y = 16, size_z = 18; MRP1, center_x = 96.035, center_y = 59.139, center_z = 57.222, size_x = 25, size_y = 25, size_z = 25; MXR1, center_x = 130.416, center_y = 129.84, center_z = 143.527, size_x = 10, size_y = 26, size_z = 22. The best molecular interactions characterized by the presence of hydrogen bonds with proteins’ key residues crucial for pump activity were identified. The revealed hit docked complexes were further visualized by BIOVIA Discovery Studio Visualized 17.2.0 (Dassault Systemes, Cedex, France).

#### 4.2.10. Colony Formation Assay

B16 cells were seeded in 96-well plates in six replicates at 100 cells/well and treated with **SM** and **SM** epoxides (0.1–0.5 µM) for 5 days under standard conditions. After 5 days, the medium was replaced with the fresh one with the same concentrations of compounds and incubated for another 5 days under standard conditions. Then, cell colonies were fixed with 4% paraformaldehyde, stained with crystal violet dye (0.1% *w*/*v*) and photographed using the iBright 1500 Imaging System (Thermo Fisher, Waltham, MA, USA). The percentage of area occupied by cell colonies was calculated using the ColonyArea ImageJ plugin [58].

#### 4.2.11. Scratch Assay

B16 and J774 cells were seeded in triplicate in 24-well plates at 3 × 10^5^ cells/well (B16) or 5 × 10^5^ cells/well (J774) and incubated under standard conditions for 24 h. Thereafter, a linear wound was generated in the monolayer with a sterile 10 µL plastic pipette tip. Cellular debris and floating cells were removed by twice washing the scratches with PBS. Then, antibiotic- and serum-free DMEM containing **SM** epoxides (0.5 and 1 µM) or **SM** (0.5 µM) was added to each well, and the cells were incubated under standard conditions for 24 h. At the time points 0 and 24 h, the scratched cell monolayers were visualized using a ZEISS Primo Vert invert microscope with a ZEISS AxioCam ERc5s camera (Carl Zeiss Microscopy GmbH, Jena, Germany). Three or four representative photos of each scratch were made. The wound closure was calculated from the normalization of the area of scratch occupied by the cells in the experimental group at 24 h to the corresponding parameter in the control group using ImageJ software (NIH, Bethesda, MD, USA).

#### 4.2.12. Adhesion Assay

B16 cells were seeded in a 96-well plate in five replicates at 10^4^ cells/well and incubated under standard conditions for 24 h. Thereafter, the cells were treated with the investigated compounds (0.2, 0.5 μM) or without them (control) for 24 h. Then, the medium was removed, 30 µL of TrypLE Express diluted with Milli-Q in a ratio of 1:1 was added to wells and the cells were incubated for 5 min at 37 °C. After that, the cells were carefully washed twice with 100 μL of cold PBS, and an MTT test was performed (as described above) to determine the relative number of cells attached to the bottom of the culture tissue-treated wells. The value of cell adhesion was determined as the ratio of the number of adhered cells in the experimental and control wells.

#### 4.2.13. Tumor Transplantation and Design of Experiment In Vivo

A metastatic model of tumor progression was generated by intravenous (i.v.) injection of B16 melanoma cells (10^6^ cells/mL) suspended in 0.2 mL of saline buffer into the lateral tail vein of C57Bl/6 mice. On day 4 after tumor transplantation, mice were assigned to groups (10 animals per group) and received intraperitoneal (i.p.) injections of **SM**, **αO-SM** and **βO-SM** at a dose of 30 mg/kg in 10% Tween-80 and **DTIC** at a dose of 40 mg/kg in water for injections. I.p. injections of 10% Tween-80 were used as the vehicle group. The treatment was carried out thrice per week. The total number of injections was eight. The mice were sacrificed on day 21 after tumor transplantation.

#### 4.2.14. Analysis of Surface Metastasis and Metastasis Inhibition Index

The lungs of the B16 melanoma-bearing mice were collected at the end of the experiment to calculate surface metastases using a binocular microscope. The inhibition of metastases development was assessed using the metastasis inhibition index (MII), calculated as described previously [59], where MII in the vehicle group was taken as 0% and MII reflecting the absence of metastases was taken as 100%.

#### 4.2.15. Toxicity Assessment

During the experiment, the general status of the animals and body weight were monitored thrice a week. At the end of the experiment, organs (lungs, livers, kidneys, spleens and hearts) were collected, and organ indexes were calculated as (organ weight/body weight) × 100%. The organ indices of B16 melanoma-bearing mice were calculated relative to the organ indices of healthy mice. Liver and kidney samples were used for subsequent histological analysis.

#### 4.2.16. Histology

For histological studies, liver and kidney specimens were fixed in 10% neutral-buffered formalin (BioVitrum, Moscow, Russia), dehydrated in ascending ethanols and xylols and embedded in HISTOMIX paraffin (BioVitrum, Russia). The paraffin sections (5 μm) were sliced on a Microm HM 355S microtome (Thermo Fisher Scientific, Waltham, MA, USA) and stained with hematoxylin and eosin. All images were examined and scanned using an Axiostar Plus microscope equipped with an Axiocam MRc5 digital camera (Zeiss, Oberkochen, Germany) at magnifications of ×400.

Morphometric analysis of liver sections was performed using a counting grid consisting of 100 testing points in a testing area equal to 3.2 × 10^6^ μm^2^ and included evaluation of the volume densities (Vv, %) of normal hepatocytes, dystrophy and necrosis as well as numerical density (Nv) of binuclear hepatocytes in the liver parenchyma. Morphometric analysis of kidney sections included an evaluation of the numbers of normal epitheliocytes of proximal tubules and epitheliocytes with dystrophic and necrotic changes, which fall into 100 counted cells. Five random fields were studied in each specimen, forming 40–50 random fields for each group of mice in total.

#### 4.2.17. Measurement of NO Production by IFNγ/LPS-Stimulated J774 Macrophages

J774 cells were seeded in 96-well plates at 10^5^ cells/well in triplicate. The plates were incubated under standard conditions for 24 h. The cells were treated with murine IFNγ (5 ng/mL), lipopolysaccharide (LPS) of E. coli (500 ng/mL) with or without the tested compounds (0.2, 0.5 µM) for 24 h. At the end of the incubation period, the level of nitrite (the main oxidation product of NO) in the culture medium was measured using the Griess reagent system (Promega, Madison, WI, USA), according to the manufacturer’s instructions.

#### 4.2.18. Carrageenan-Induced Peritonitis

Balb/C mice were intraperitoneally (i.p.) pretreated with **SM**, **αO-SM** and **βO-SM** at a dose of 30 mg/kg in 10% Tween-80 and dexamethasone at a dose of 1 mg/kg 1 h before peritonitis induction by 1% carrageenan (i.p.); 10% Tween-80 i.p. was used as the vehicle group. Four hours after peritonitis induction, mice were sacrificed, and the peritoneal cavities were washed with 2 mL of heparinized cold saline buffer to obtain peritoneal exudates. The collected samples were centrifuged (2500 rpm, 5 min, 4 °C), the cell pellets were resuspended in 50 μL of PBS and total leukocyte counts were performed using a Neubauer chamber by optical microscopy after diluting the peritoneal fluids with Turk solution (1:20). To determine the differential leukocyte counts, peritoneal cells were placed onto slides, stained with azur-eosin by the Romanovsky–Giemsa and examined by optical microscopy. The results were expressed as the number of total leukocytes (×10^5^/mL) and the ratio of neutrophils, lymphocytes and monocytes (%). The supernatants were collected to assess the levels of pro-inflammatory cytokine TNF-α using a mouse TNF-α ELISA kit (Thermo Scientific, Rockford, IL, USA) according to the manufacturer’s instructions.

#### 4.2.19. Statistical Analysis

Statistical analysis was performed using the Microsoft Excel program, assuming a significant level of changes at *p* < 0.05. The two-tailed unpaired t-test assessed the statistical significance of the experimental and control groups.

## Data Availability

Not applicable.

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
