# Peer review of "Novel Epoxides of Soloxolone Methyl: An Effect of the Formation of Oxirane Ring and Stereoisomerism on Cytotoxic Profile, Anti-Metastatic and Anti-Inflammatory Activities In Vitro and In Vivo"

_ijms, 2022, doi:10.3390/ijms23116214_

Round 1
Reviewer 1 Report
The authors synthesized two novel epoxide of soloxolone methyl derivatives. This manuscript is very well written and all experiments good docummented.
I think that it should be accepted for publication after minor revisions.
1. The conclusion is to long and the first sentence is not informative. Please check the number of words required by the journal.
2. What is with solubility and stability of the new compounds.
Author Response
Dear Reviewer #1,
We are very grateful to you for the valuable comments that helped us to improve the manuscript. We revised the manuscript according to your comments and, please, let us respond to your remarks.
- The conclusion is to long and the first sentence is not informative. Please check the number of words required by the journal.
Authors: Corrected. Indeed, the Conclusion section contained unnecessary extra information about IC50 values of the compounds and used cellular models. This section has been markedly shortened. Please, see lines 543-561.
- What is with solubility and stability of the new compounds.
Authors:
Solubility. The solubility of the resulting compounds is comparable to that of the parent compound SM (Stock solutions of triterpenoids at 10 mM in DMSO).
Stability as a dry solid. The resulting new epoxides are quite stable in dry form at room temperature. To date, the substances are stored for about 12 months without any traces of decomposition. Stability in solutions: 1) the substances do not undergo changes when kept in the acetonitrile-water (3:1 by volume) system at room temperature for 14 days (not measured further) (analysis by HPLC); 2) the substances are also stable in chloroform and DMSO for 4 days at 4°C (not measured further) (according to NMR spectra for solutions in CDCl3 and DMSO-d6, respectively).
Beside this, additional characteristics of the compounds were added in Sections 4.1.2 and 4.1.3 (lines 600-601, 627-628; marked yellow) and NMR 1H and 13C and HRMS data of αO-SM and βO-SM were added as Supplementary file.
We hope that corrected version of the manuscript will be acceptable for publication in the International Journal of Molecular Sciences.
Sincerely,
On behalf of all authors,
Dr. Andrey Markov

Reviewer 2 Report
This study investigated whether the formation of an epoxy structure on SM ring A could impact the anti-cancer and anti-inflammatory effects of the parent substance, SM. The epoxy structure sounds pharmacologically active. To show the anti-cancer and anti-inflammatory effects, the authors executed various appropriate experiments and used various malignant cells. However it seems that novel epoxides did not have big differences with the parent substance in terms of anti-cancer and anti-inflammatory profiles. The notable difference was restoring of the leukocyte subpopulation, which made sure the effect of the cyanoenone pharmacophore group of SM. The story is somewhat negative but theoretical and well according to the experimental results.
Minor points
L93 : “Moreover, it is remains…” might be “Moreover, it remains…”.
L172: How to calculate “Selectivity index” and the meaning of it should be explained in the text around “Table2”.
L266: What is “snugly”?
Fig.4F: What is “organ index”? It should be explained.
L529, L532: Two (Figure5H) might be (Figure 5G).
Author Response
Dear Reviewer #2,
We are genuinely thankful to you for your careful analysis of our manuscript and highly valuable remarks. We revised the manuscript according to your comments and, please, let us respond to your comments.
- L93 : “Moreover, it is remains…” might be “Moreover, it remains…”.
Authors: Corrected.
- L172: How to calculate “Selectivity index” and the meaning of it should be explained in the text around “Table2”.
Authors: Selectivity index of evaluated compounds was calculated as the ratio of their IC50 values on non-transformed hFF3 fibroblasts to their IC50 values on corresponding tumor cell line. The explanation of SI was added to the footnote of Table 2 (please, see lines 173-174).
- What is “snugly”?
Authors: “Snugly fit” means that compound formed optimal docking complex with transmembrane domain of ABC-transporters, forming the network of bonds with key amino acid residues. To make this phrase more understandable “snugly” was replaced with “properly” (please, see line 269).
- What is “organ index”? It should be explained.
Authors: The organ index is the ratio of the organ weight to the body weight of the mice. The explanation of “organ index” was added to the caption of Figure 4F (please, see lines 409-410).
- L529, L532: Two (Figure5H) might be (Figure 5G).
Authors: Corrected. Thanks a lot for the careful analysis of our manuscript! It was our disappointing misprint.
Beside this, additional characteristics of the compounds were added in Sections 4.1.2 and 4.1.3 (lines 600-601, 627-628; marked yellow) and NMR 1H and 13C and HRMS data of αO-SM and βO-SM were added as Supplementary file.
We hope that this version of the manuscript will be acceptable for publication.
Thank you very much!
Sincerely,
On behalf of all authors,
Dr. Andrey Markov
